# Simple Fully Connected Network for Composing Word Embeddings from Characters

## Abstract

This work introduces a simple network for producing character aware word embeddings. Position agnostic and position aware character embeddings are combined to produce an embedding vector for each word. The learned word representations are shown to be very sparse and facilitate improved results on language modeling tasks, despite using markedly fewer parameters, and without the need to apply dropout. A final experiment suggests that weight sharing contributes to sparsity, increases performance, and prevents overfitting.

## 1 Introduction

When processing text for Natural Language Processing (NLP), one important decision to make is how to represent the words for a given model or system. For many tasks tackled by deep learning such as language modeling, language understanding, and translation, the use of *word embeddings* has become the standard approach. (Zaremba et al., 2014) (Cheng et al., 2016) (Bahdanau et al., 2014) This is in part due to their ability to represent complex syntactic and semantic relationships between words as spatial relationships within the embedding dimensions (Mikolov et al., 2013).

Embeddings are generally implemented as a lookup table for computational efficiency. However for those unfamiliar with their use it may be beneficial to formulate them as the output of the first layer in a neural network. This is true for a layer that has one-hot feature vectors as inputs, no bias, and no activation function.

For a given one-hot feature vector $x$, the activations of such a layer can be computed by $xW$, which is equivalent to selecting the row $W_i$ of the weight matrix, where $x_i == 1$. The weight matrix or embedding lookup matrix can then be optimized via typical techniques such as gradient descent, including from subsequent layers of a DNN through back propagation. (Rumelhart et al., 1988)

For word embeddings, the basic approach is to assign a unique vector of trainable parameters to each word in a vocabulary. These vectors are referred to in this paper as token embeddings. Token embeddings present a number of limitations. For example, any out-of-vocabulary word cannot be represented except as a pre-defined $< UNK >$ token. A corollary of this is that the number of embeddings (and therefore trainable parameters) grows linearly with the size of the vocabulary. Furthermore, characters are ignored, meaning that potentially useful morphological information is thrown out.

To get around these limitations, researchers have explored building word embeddings from lower level character representations. A variety of techniques have been presented, including the use of feedforward multi layer perceptrons (MLPs) (Chen et al., 2015), convolutional neural networks (CNNs) (Józefowicz et al., 2016) (Kim et al., 2015), and recurrent neural networks (RNNs) (Luong & Manning, 2016). These character level representations of words have the advantage over token embeddings of allowing an open vocabulary, usually having fewer parameters, and improving performance by making use of information available in sub-word level features.

The most successful approaches for building word embeddings from characters use CNNs. (Józefowicz et al., 2016) However, the architecture of CNNs is designed to identify position-invariant features, not the specific ordering of characters that make up a word's spelling. Here we ask whether such ordering is a valuable source of information.

A number of convolutional features of varying size can be used to capture some ordering, for example within each feature independently. However as the vocabulary is expanded, the number convolutional features must be increased to compensate (Józefowicz et al., 2016). Once convolution is performed, the used of a deep highway network, as introduced by Srivastava et al. (2015), is then needed to produce the final word embedding.

The current study presents a simple fully connected architecture for combining characters. In this framework, each character is represented both by position-agnostic character embeddings and position-aware character embeddings, which we call spelling embeddings. The combination of the two allows the model to learn both position invariant features and positional features. A word embedding is then constructed by combining both the character and spelling embeddings of the word, for example by summing or by averaging them together. The resulting vector is then passed through a nonlinear MLP that combines the character and spelling information to produce the final word embedding.

This MLP, along with the spelling and character embeddings, were trained via gradient descent as inputs to a Recurrent Neural Network (RNN) being trained for a language modeling task. Results show that including the spelling information facilitates improvement over token embeddings despite requiring far fewer parameters. Without the position information, character embeddings alone are not sufficient in this fully connected architecture.

An analysis of the learned representations at the word embedding level shows much greater sparsity for spelling embeddings than for token embeddings, and demonstrates some of the negative impacts of dropout on the representations. Finally, we compare token based models with a fully connected layer of shared weights to raw token embeddings with no weight sharing. Passing the token embeddings through a layer of shared weights is shown to drastically increase representation sparsity and prevent overfitting. Given that the character and spelling weights are heavily shared among word embeddings, this is presented as possible explanation for the spelling aware model's robustness against overfitting.

## 2 PREVIOUS WORK

Many architectures have been explored for composing word embeddings from lower level features, including the use of recurrent neural networks (Luong & Manning, 2016), (Ling et al., 2015) convolutional networks (Kim et al., 2015), (Santos & Zadrozny, 2014), (Józefowicz et al., 2016), character n-grams (Bojanowski et al., 2016), as well as combinations of word tokens with morphological features (Botha & Blunsom, 2014), (Chen et al., 2015).

One such architecture is to enhance token based word embeddings of Chinese words by including character embeddings (Chen et al., 2015). Multiple approaches were explored, the simplest of which was to embed characters and build a word embedding by combining a traditional token embedding with the average of the embeddings for each character in the word:

$$e(i) = T_i + \frac{1}{L_i} \sum_{i=0}^{L_i} c_j$$

Where $e(i)$ is the character enhanced embedding for word $i$, T is the token embedding lookup table, $T_i$ is the token embedding vector for the word, $c_j$ is the character embedding vector for the $j$th letter of the word, and $L_i$ is the total number of letters in the word.

There are a number of drawbacks with this approach. First, character ordering is not taken into account so the token embeddings are needed to ensure uniqueness. Second, the character embeddings were not included for words that were pre-screened for ambiguous or misleading character information, which requires a manual or heuristic pre-processing step. Finally, simply averaging the character embeddings doesnt provide an opportunity to build richer non-linear combinations such as would be possible with an MLP.

Convolution neural networks (CNNs) have also been used to create word embeddings from character representations. (Józefowicz et al., 2016) Their character aware CNN architecture was based on a previous publication by Kim et al. (2015), but used more convolution features (4096) to cope with larger datasets. This approach was found to give state of the art results when applied to language

modeling with the popular One Billion Word Benchmark, despite using far fewer parameters than a traditional token embedding model. The use of a fully connected network with explicit positional information was not reported on.

The inclusion of positional information can be handled in a variety of ways. An interesting method not explored in this work is provided by Vaswani et al. (2017), who combine positional information with each symbol in the form of unlearned $sin$ and $cosine$ dependant functions of varying frequencies. These functions produce repeating waveforms that allow their model to capture information about relative positions. This differs from the current study which uses learned, explicit and distinct representations for each position of each character.

## 3 METHODS

### 3.1 LANGUAGE MODELING

The task in language modeling is to assign probabilities to sentences or sequences of words. That is, we want to model the probability of of the next word in a sequence conditional on the ordered sequence of all previous words.

$$p(w_t = i | w_{t-1}, w_{t-2}, ... w_0)$$

This was accomplished with the use of an RNN which produces a context vector $v$ from previous words. The RNN was implemented with Gated Recurrent Units (GRUs), which we denote here by the function $g()$ for simplicity. (Cho et al., 2014) A fully connected layer with weights $W^{(s)}$ and biases $b^{(s)}$ was used to project the GRU's output, $v$, to the target vocabulary. A softmax activation was then applied to produce a valid probability distribution, $q$, over the vocabulary:

$$v_t = g(v_{t-1})$$
$$s_t = v_t W^{(s)} + b^{(s)}$$
$$q(w_t = i) | v_t) = \frac{e^{s_{ti}}}{\sum_{j=0}^{V} e^{s_{tj}}}$$

Gradients were computed from the cross entropy between the softmax layer and expected next word in the sequence. Each batch contained sequences with a fixed length, and each sequence followed from the previous batch. Gradients were time truncated to the fixed sequence length. Gradients were also clipped to a maximum global norm to prevent them from exploding. (Pascanu et al., 2012) The initial states of the GRU were only reset at the beginning of each epoch. Dropout was applied to the outputs of each RNN layer in order to regularize the RNN weights of the models. (Srivastava et al., 2014) For the embedding layers, two dropout configurations were compared. The first applied dropout to the final word embedding layer and the second did not.

### 3.2 DATASETS

Two datasets were evaluated. The first is a relatively small dataset consisting of the novels of *The Wheel of Time*, by Jordan & Sanderson (1990-2013). It has vocabulary of 34,594 words and was split into train/test partitions with 5,007,362 and 444,576 words respectively.

The second dataset is a subset of works from Project Gutenberg Canada, which is a collection of writings that are in the Canadian public domain. This larger dataset has a vocabulary of 205,027 words and was split into train/test partitions with 63,319,830 and 7,136,409 words respectively. This is about 10% the size of the popular One Billion Word Benchmark.

Both datasets were pre-processed as follows. All texts were lower-cased and separated by whitespace into words. Each word was then parsed again so that any stream of consecutive alphabetical characters were considered a token, and any stream of non-alphabetical characters were considered a token. Converting the vocabulary to lowercase removes potentially valuable information, and was done only to reduce the vocabulary size. This allowed for a speed up in experimentation and hyper-parameter tuning, as well as to fit larger models on available hardware.

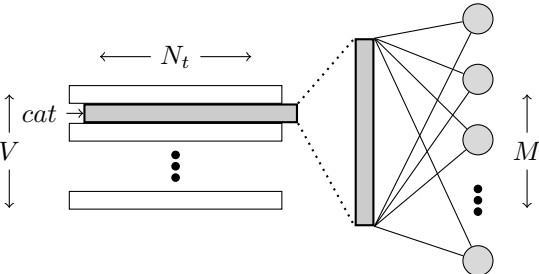

Figure 1: Graphical representation of the process for producing the token embedding for the word *cat*. The token embedding for the word corresponds to single row of the embeddings matrix, which is then presented to a fully connected layer.

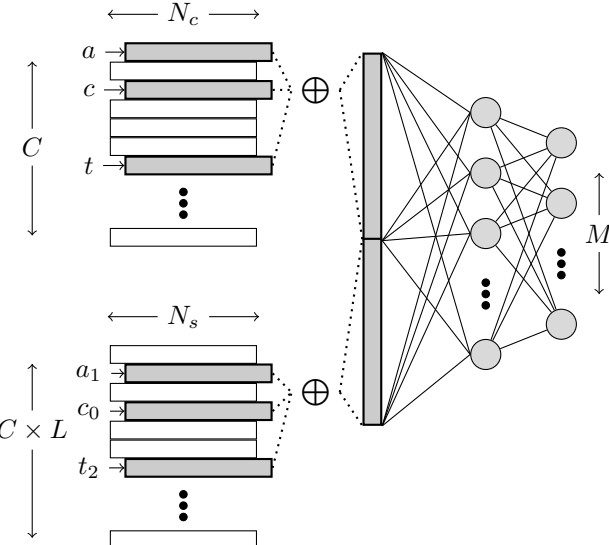

Figure 2: A graphical representation of the process for embedding the word *cat* using spelling embeddings. The position aware and position agnostic embeddings are selected and averaged from their respective matrices. The results are then concatenated before being presented to an MLP.

### 3.3 EMBEDDINGS

The token embeddings consist of a $V \times N$ lookup table of trainable parameters followed by a fully connected layer with rectified linear units (ReLUs). A graphical representation is provided in figure 1. Given a lookup table $T$, word index $i$, and a fully connected layer with matrix $W^{(t)}$ and bias vector $b^{(t)}$, the embedding funtion is:

$$e^{token}(i) = relu(T_i W^{(t)} + b^{(t)})$$

An additional configuration for the tokens, referred to as raw token embeddings, was investigated with the larger dataset. These were simply presented directly the the RNN, rather than passed through a fully connected layer first. Hence:

$$e^{rawtoken}(i) = T_i$$

As shown in figure 2, the spelling embeddings are built up from the characters in the word as follows. Two lookup tables are used. One contains position agnostic character embeddings and is of size $C \times N_c$. The other contains positional character embeddings and is of size $C \times L \times N_s$. Where $C$ is the number of characters, $L$ is the maximum word length, $N_c$ is the size of the embedding

dimension for position agnostic character embeddings, and $N_s$ is the embedding dimension for spelling embeddings.

To embed a word, the embeddings for the characters in the word are first pulled from each table separately and averaged. The resulting vectors from these averages are then concatenated together to produce a vector of dimensionality $N_c + N_s$. This vector is then used as input to an MLP with two ReLU layers.

We denote the lookup tables for the position aware and position agnostic character embeddings as $U$ and $V$, respectively. Then for a word indexed by $i$, the vector $w^{(i)}$ contains the indices corresponding to the position agnostic characters of that word. Then if $L^{(i)}$ is the length of the word and $j$ indexes the character position, we formulate the concatenation ($\|$) of the position aware and position agnostic character representations of the word as:

$$ f(i) = \frac{1}{L^{(i)}} \sum_{j=1}^{L^{(i)}} U_{w_j^{(i)}} \| \frac{1}{L^{(i)}} \sum_{j=1}^{L^{(i)}} V_{jw_j^{(i)}} $$

The models were also run without the position aware spelling embeddings in order to determine the value of this information for the task. All embedding methods presented have a final embedding layer with dimensionality $M$, in order to ensure that the language model is given an equal capacity for information about incoming words.

Table 1: Number of Parameters to Embed Vocabulary

|  |  | The Wheel of Time | Gutenberg |
|---|---|---|---|
| spelling |  | $12,304,800$ | $13,020,000$ |
| token |  | $13,997,600$ | $82,170,800$ |

The number of parameters required to embed the entire vocabulary was controlled in order to prevent spelling embeddings from gaining an unfair advantage over tokens. This was accomplished by limiting the number of nodes in each layer. One of the main benefits of spelling embeddings is that the number of parameters does not grow necessarily with the vocabulary size as it does with token embeddings. The number of parameters needed to embed the vocabulary using token embeddings is computed by $(V \times N_t) + (N_t \times M)$. The dominant term is generally the vocabulary size, $V$, which is much larger than the embedding dimension.

For spelling embeddings an upper bound is considered because not all characters must appear in all possible word positions. This is computed by:

$$ (N_c \times C) + (N_s \times C \times L) + ((N_s + N_c) \times M') + (M' \times M) $$

where $M'$ is the size of the fully connected layer placed between the character embeddings and the final embedding layer.

Table 1 shows the specific values for the number of parameters used in our experiments. The spelling embeddings do not use significantly more parameters for the larger dataset than for the smaller because they depend on the number of characters and the lengths of words rather than on the number of words.

## 4 RESULTS

On the larger Gutenberg dataset, spelling embeddings outperform token embeddings despite using far fewer parameters to embed each word ($\sim 13M$ vs $\sim 82M$). On the smaller *Wheel of Time* dataset they are on par with token embeddings. Position agnostic character alone embeddings perform worse than tokens. Performance curves are plotted in figure 3. Final performance of each model is listed in table 2. The token embeddings overfit the training data on the *Wheel of Time* dataset. On the Gutenberg dataset, only the raw token embeddings exhibited overfitting.

Figure 4 provides a visual inspection of three randomly chosen word embeddings across each of the experiments done on the Gutenberg dataset. These visualizations show that the activations of

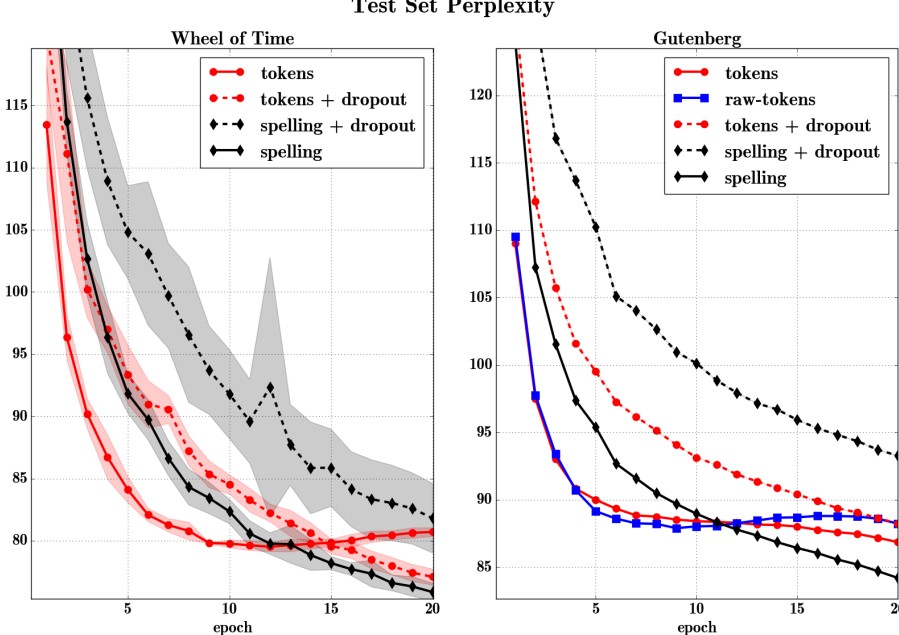

Figure 3: Per-word perplexity on each test set after each training epoch. Comparing both the embedding style (token vs spelling) and whether or not dropout was included on the final embedding layer. The Wheel of Time plot shows the mean value over 5 runs with the standard deviation represented by the shaded regions. The Gutenberg plot shows an additional curve for the raw token embeddings

Table 2: Final Per-Word Perplexity

|  | The Wheel of Time | Gutenberg |
|---|---|---|
| spelling + dropout | $81.81 \pm 2.79$ | 93.26 |
| spelling | **$75.95 \pm 0.75$** | **84.19** |
| position agnostic chars | $77.35 \pm 0.75$ | 94.12 (10 epochs) |
| token + dropout | $77.09 \pm 0.65$ | 86.86 |
| token | $80.70 \pm 0.36$ | 88.23 |

spelling embeddings are far more sparse than those of token embedding. Raw token embeddings exhibit the least amount of sparsity.

To get a more comprehensive view of sparsity, the Gini coefficient was applied to the embeddings of the entire vocabulary for each model run on the Gutenberg dataset. (Gini, 1912) The Gini coefficient was chosen as a measure of sparsity because it has been shown to be robust under a number of metrics. (Hurley & Rickard, 2008) Figure 5 shows the distribution of sparsity across the vocabulary as measured by the Gini coefficient. Raw token embeddings are the least sparse. Token embeddings passed through a fully connected layer increase dramatically in sparsity, followed by the spelling embeddings which are the most sparse. Sparsity is also affected by dropout. Whereas dropout results in greater sparsity for the majority of the token embeddings, it causes a few to lose all sparsity and become completely homogeneous. Dropout also has this homogenizing effect on some of the spelling embeddings.

## 5 DISCUSSION

This work shows that a simple fully connected network is able to produce character aware word embeddings that outperform traditional token embeddings. The architecture is relatively simple

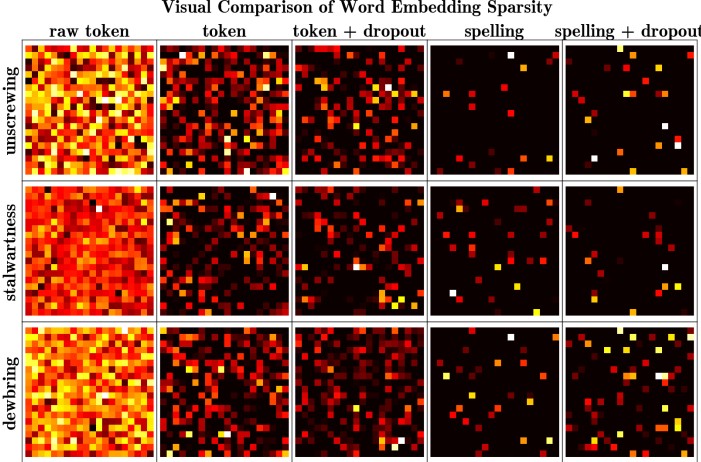

Figure 4: For each experiment, displaying the word embeddings of 3 randomly selected words. The embeddings have 400 nodes which have been reshaped to $20 \times 20$ and visualized as a heat map. White indicates the most highly activated node, black the least.

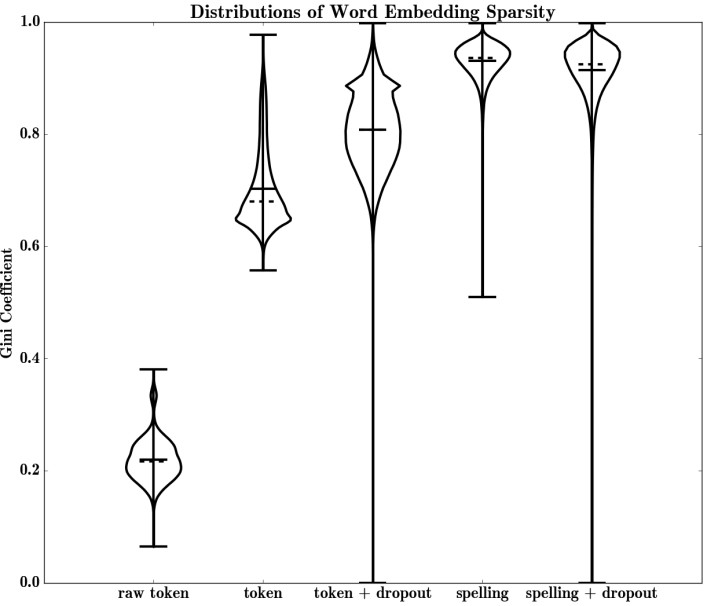

Figure 5: For each experiment, shows the distribution of sparsity measurements over all trained word embeddings in the vocabulary.

compared to previous approaches that use CNNs or RNNs to combine character information. This work lacks a direct comparison to these other character aware methods, which is an obvious direction for future work.

Investigation into the word embeddings produced by the presented architectures reveal a number of interesting properties. Spelling embeddings are especially resistant to overfitting compared to token embeddings, and are also significantly more sparse in their activations. Furthermore, dropout is

shown to have some negative impacts on the word representations, and weight sharing is presented as a better way to regularize word embeddings.

Spelling embeddings exhibit the most weight sharing, because each character embedding is shared across many words in the vocabulary. This may be a contributing factor to their increased sparsity and resistance to overfitting. Additional evidence for this is provided in the comparison of raw token embeddings to those passed through a fully connected layer.

Whereas raw token embeddings share none of their weights with other words in the vocabulary, token embeddings passed through a fully connected layer share all the weights in that layer across the entire vocabulary. Not only do token embeddings enjoy increased resistance to overfitting when passed though a shared weight layer, they also become drastically more sparse. Whereas dropout is a popular technique for regularization in NLP, it can have a negative impact on the word embeddings, causing some of them to gain a Gini coefficient of $0$. This suggests that these particular words have completely homogeneous representations and are indistinguishable from each other.

On the smaller dataset the number of *shared* parameters in the fully connected layer of the token embeddings is large compared to the vocabulary size. In this case, dropout is needed to prevent overfitting. On the larger dataset, the number of *shared* parameters is much smaller relative to the vocabulary size. In this case dropout is not needed for the token embeddings and actually hinders them. The spelling embeddings perform worse with dropout on both datasets.

The architecture presented here should be compared to the state of the art character CNN results obtained on the One Billion Word benchmark. (Józefowicz et al., 2016) Also, whereas a number hyper-parameters governing the number and size of the layers were tried before the ones presented in this paper were found, other techniques such as highway networks (Srivastava et al., 2015) have not yet been investigated.

Furthermore, extending the concept of character aware word embeddings to the output softmax layer is another open area of research that has been tried with character CNNs (Józefowicz et al., 2016), but not to our knowledge with a spelling network as presented in this work.

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
