# OpenReview forum: "A Simple Fully Connected Network for Composing Word Embeddings from Characters"
_ICLR.cc/2018/Conference — Reject_

### Official Review · AnonReviewer3 · 2017-11-25
**The authors implement a model to obtain an embedding for a word given its characters, but fail to compare to relevant previous work or even any published results, making the validity of their claims difficult to asses.**

**Rating:** 3
**Confidence:** 4

**Review:**

The authors propose a neural network architecture which takes the characters of a word as input along with their positions, and output a word embedding. They then use these as inputs to a GRU language model, which is evaluated on two medium size data sets made from a series of novels and the Project Gutenberg Canada books respectively.

While the idea has merit, the experimental protocol is too flawed to draw any reliable conclusions. Why use Wheel of Time, which is not in the public domain, rather than e.g. text8? Why not train the model to convergence (Figure 3)? Do the learned embeddings exhibit any morphological significance, or does the model only serve a regularization purpose?

As for the model itself: are the position agnostic character embeddings actually helpful in the spelling model? Does the model have the expressivity to learn the same embeddings as a look-up table?

The authors are also missing a significant amount of relevant literature on the topic of building word embeddings from characters, for example:
Finding Function in Form: Compositional Character Models for Open Vocabulary Word Representation, Ling et al., 2015
Enriching Word Vectors with Subword Information, Bojanowski et al. 2017
Compositional Morphology for Word Representations and Language Modelling, Botha and Blunsom 2014

Pros:
- Valid idea

Cons:
- Too many missing references
- Some modeling choices lack justification
- Experiments do not provide meaningful comparisons and are not reproducible

---

### Official Review · AnonReviewer1 · 2017-11-27
**Position aware character embeddings can be sparse and are less prone to overfitting on language modeling.**

**Rating:** 5
**Confidence:** 4

**Review:**

The paper uses both position agnostic and position aware embeddings for tokens in a language modeling task. To obtain token embeddings, they concatenate two embeddings: the sum of character embeddings and the sum of (character, position) embeddings, the former being position agnostic and the latter being position aware. In a language modeling task, they find that using a combination of both improves perplexity over the standard token embedding baseline with fewer parameters.

The paper shows that the character embeddings are more sparse, measured with the Gini coefficient, than token embeddings and are more robust to overfitting. They also find that while dropout increases overall sparsity, it makes a few tokens homogenous. The paper does not give a crisp answer to why such sparsity patterns are observed.

The paper falls a bit  short both empirically and technically. While their technique is interesting, they do not compare it to the baseline of using convolutions over characters. More empirical evidence is needed for the technique to be adopted by the community.  On the theory side, they should dig deeper into the reasons for sparsity and how it might help to train better models.

If the papers shows that the approach can work well in machine translation or language modeling of morphologically rich languages, it might encourage practitioners to use the technique.

---

### Official Review · AnonReviewer2 · 2017-11-27
**Another model for learning to embed words as a function of their characters**

**Rating:** 4
**Confidence:** 5

**Review:**

This paper presents a new model for composing representations of characters into word embeddings. The starting point of their argument is to include position-specific embeddings of characters rather than just position-independent characters. By adding together position-specific vectors, reasonable results are obtained.

This is an interesting result, but I have a few recommendations to improve the paper.
1) It is a bit hard to assess since it is not evaluated on a standard datasets. There are a number standard datasets for open vocabulary language modeling. E.g., the MWC corpus (http://k-kawakami.com/research/mwc), or even the Penn Treebank (although it is conventionally modeled in closed vocabulary form).
2) There are many existing models for composing characters into words. In addition to those cited in the paper, see the citations listed below. Comparison with those is crucial in a paper like this.
3) Since the predictions are done at the word type level, it is unclear how vocabulary set of the corpus is determined, and what is done with OOV word types at test time (while it is possible to condition on them using the technique in the paper, it is not possible to use this technique for generation).
4) The analysis is interesting, but a more intuitive explanation would be to show nearest neighbor plots.

Some missing citations:

Composing characters into words:

dos Santos and Zadrozny. (2014 ICML) http://proceedings.mlr.press/v32/santos14.pdf
Ling et al. (2015 EMNLP) Finding Function in Form. https://arxiv.org/abs/1508.02096

Additionally, using explicit positional features in modeling language has been used:
Vaswani et al. (2017) Attention is all you need https://arxiv.org/abs/1706.03762
and a variety of other sources.

---

### Decision · Program_Chairs · 2018-01-29
**ICLR 2018 Conference Acceptance Decision**

**Decision:**

Reject

**Comment:**

The paper presents yet another approach for modeling words based on their characters. Unfortunately the authors do not compare properly to previous approaches and the idea is very incremental.